# Evolution of the Habits of Physical Activity and Television Viewing in Spanish Children and Pre-Adolescents between 1997 and 2017

**DOI:** 10.3390/ijerph17186836

**Published:** 2020-09-18

**Authors:** Jose L. García-Soidán, Raquel Leirós-Rodríguez, Vicente Romo-Pérez, Víctor Arufe-Giráldez

**Affiliations:** 1Faculty of Education and Sport Sciences, University of Vigo, Campus a Xunqueira, s/n, 36005 Pontevedra, Spain; jlsoidan@uvigo.es (J.L.G.-S.); vicente@uvigo.es (V.R.-P.); 2Department of Nursing and Physical Therapy, Faculty of Health Sciences, University of León, Ave. Astorga, 15, 24401 Ponferrada, Spain; 3Faculty of Education, University of Coruña, Campus de Elviña, s/n, 15008 Coruña, Spain; v.arufe@udc.es

**Keywords:** lifestyle, sedentary behavior, prevention and control, child behavior, child health, screen time, motor activity

## Abstract

Background: Promoting healthy lifestyles in children, has become a priority for public health institutions. However, electronic devices with screens encourage sedentary behaviors. The aim of this study was to analyze the evolution of the habits of physical activity practice and television watching in a cohort of 20 years of research in Spanish children. Methods: A cross-sectional, observational study was based on data from the Spain National Health Surveys between 1997 and 2017 (*N* = 11,444). The dependent variables considered were the frequency with which the minor practiced physical activity (PA) in his/her spare time, the daily TV viewing habit, and the daily hours of TV viewing. Results: The children who practiced physical activity daily has decreased 7.3% throughout the study period. The proportion of children who watched the television daily for more hours increased significantly (6.3%). Such increase was constant throughout the years, and the analysis by sex showed that the proportion of boys who watched television for more than three hours per day increased three percent, and that of girls increased fourfold. Conclusions: The habits of physical activity practice and television viewing have changed towards sedentary lifestyle. Particularly, the girls and the children between 12 and 14 years showed the most sedentary behavior. Public health policies must consider the differences between sexes in order for such interventions to be effective in the population of pre-adolescents, in general, and girls, in particular.

## 1. Introduction

In the last decades, lifestyles have changed considerably worldwide. In western and developed countries, the time spent in sedentary activities, such as watching television (TV), has increased exponentially [1,2]. This alteration in daily habits is related to the increased rate and prevalence of obesity, cardiovascular diseases, cancer, diabetes and, in general, to more comorbidities, healthcare expenses, and premature mortality [3,4].

In the specific study of the children population, the number of five-year-old children with obesity are expected to be 41 million by the year 2030 [5]. This datum is especially relevant, since the obesity established during childhood is more likely to prevail for the entire life of the individual compared to the obesity acquired during adulthood [5,6]. The origin of the increase in the weight and body mass index (BMI) and, consequently, of the overweight is multifactorial and behind it lies a complex system of interconnected factors [7,8]. In the case of children, these factors include daily energy consumption and expenditure, energy patterns physical activity (PA), and genetic, environmental factors, values socio-cultural, social class, economic factors, and stress [9]. For all this, promoting active and healthy lifestyles in the general population, especially in children, has become a priority for public health institutions [10,11].

The practice of PA in children has multiple beneficial consequences. One of the most relevant benefits is the secretion of neurotransmitters and hormones that favor the intellectual and cognitive development by generating new neurons and multiplying and strengthening the neural connections between the brain areas related to memory and learning [12,13]. These mechanisms, which result from the practice of PA regardless of age, are especially relevant during growth, since physical exercise improves basic intellectual competences, such as emotional control, memory and the capacity to adapt to different tasks and environments, which are fundamental for an optimal integral development as individuals [14,15]. Moreover, at the physical level, PA improves the general health state and body composition and it promotes the good cardiovascular, respiratory, and metabolic function [16,17]. However, a recent research has indicated that the percentage of children who do not perform PA even once a week reaches 52.9% [18].

However, electronic devices with screens, such as TVs, computers, tablets, and mobile phones, encourage sedentary behaviors and are associated with the higher rate of emotional problems (such as anxiety, depression, and lower self-control in children) [19,20]. Despite the increased amount of multimedia devices with screens present in the household, the TV is still the dominant device in terms of time spent by children in front of a screen [21,22]. In fact, in the year 2019, 9 out of 10 Spanish homes had access to the Internet, 8 out of 10 had at least one computer, 5.6 out of 10 had at least one tablet, 9.8 out of 10 had at least one mobile phone, and 9.9 out of 10 had at least one TV (Spanish Institute of Statistic) [23]. Watching TV implies, fundamentally, adopting a physically inactive position and, consequently, a low energy expenditure. Therefore, a longer time spent watching TV during childhood has been associated, in multiple cases, with overweight and obesity, as well as with disturbed-sleep behaviors and improper dietary habits [24,25,26]. However, Mutz, Roberts, and van Vuuren [27] concluded that the displacement mechanism is asymmetric in nature; that is, although increases in television viewing force out some other activities, decreases in television viewing do not result in parallel increases in levels of any of these activities. Although this pattern of findings was most pronounced in the case of radio use and movie attendance [27]. Thus, how and how much television “competes” for children’s free time has yet to be established and, if so, it is not known whether it does it differently in boys and girls. 

Considering the above mentioned, the aim of this study was to analyze the evolution of the habits of PA practice and TV watching in a cohort of 20 years of research (1997–2017) in Spanish minors, in general, and children, specifically, in order to determine whether there is a relationship between these two activities, the BMI of the children, and if there are differences in these habits between boys and girls. The working hypotheses stated that the habits of Spanish children were altered, that such alteration was different according to sex and that these two activities were related.

## 2. Materials and Methods

### 2.1. Study Design and Sample

This cross-sectional, observational study was based on data from the NHS (National Health Surveys) about minors, carried out by the Spanish Institute of Statistics and the Spanish Ministry of Health, Consumption, and Social Welfare. All surveys were conducted through systematic sampling, with the equiprobability of being selected within their corresponding census section in the entire Spanish territory. Specifically, personal interviews were conducted by telephone with representative samples of the Spanish society in the year 1997 (*n* = 2000), 2001 (*n* = 5280), 2006 (*n* = 9122), 2011 (*n* = 7214), and 2017 (*n* = 6106).

In the present study, we used the data from 29,722 surveys completed by adults who lived with at least one minor of up to 15 years of age. To carry out this investigation, those who lived with children under six years of age were excluded, since the quantification of the PA practiced by minors of this age without objective monitoring or direct observation is not reliable [28]. The application of such selection criteria generated a sample of 11,444 minors, with a sex distribution of 51.6% men and 48.4% women and an average age of 10.3 ± 3 years. 

### 2.2. Study Variables

The National Health Survey is a statistical state investigation, designed with the purpose of obtaining data on people’s health status, use of health services, prevention, and risk factors. All natural and legal persons that provide data must answer truthfully, accurately, completely, and within the term to the questions ordered in the due form by the statistical services (Article 10.2 of the Law of Public Statistical Function) [29]. The following dependent variables were considered:
(1)Frequency with which the minor practiced PA in his/her spare time, measured through the following question: Which one of these options best describes the frequency with which the minor practices some physical activity in his/her spare time? The response options were (a) does not exercise, (b) practices some PA or sports activity occasionally (less than once per month), (c) practices some PA or sports activity regularly (several times per month), and (d) practices some PA or sports activity frequently (several times per week).(2)Daily TV viewing habit, measured through the following dichotomous question: “Does the minor watch TV daily?” The response options were “yes” or “no”.(3)Daily hours of TV viewing, measured through the following question: "On average, how much time does the minor spend watching TV every day?" The response options were (a) never, (b) less than one hour, (c) between one and two hours, (d) between two and three hours, and (e) more than three hours.

The independent variables were (1) sex; (2) age (years); (3) weight (kg); and (4) BMI (kg/m^2^), which was calculated from the weight and height stated by the adults in the interviews and classified as underweight, normal weight, overweight, or obese, based on the guidelines of the World Obesity Federation [30], categorizing as underweight any BMI value that does not reach the minimum healthy level for the corresponding sex and age. Overweight and obesity in minors were determined by +1 and +2 standard deviation, respectively, which is equivalent to 25 kg/m^2^ and 30 kg/m^2^, respectively, in adults [31]. The reliability and validity of the anthropometric measurements stated by parents has been proven and accepted in previous research [32].

### 2.3. Statistical Analysis

For the analysis of the results, the sample of minors was divided into three age groups: G1, 6–8 years (*n* = 3098), G2, 9–11 years (*n* = 3749), and G3, 12–14 years (*n* = 4597). This division responds to the need to divide the sample equitably and that the older children (those who in previous research are usually included as “pre-adolescents”) were all in the same group [33,34].

Descriptive measures were used to characterize the sample (frequency, percentage, mean, and standard deviation). The ANOVA test with the Bonferroni correction was used to determine the differences between the age groups and between the samples of the different analyzed surveys. To compare the variables according to gender, Student’s *t*-test for independent samples (continuous data) and the Chi-square test were used (categorical data).

Furthermore, multinomial logistic regressions were conducted to identify which factors were associated with more PA practice (reference condition). The independent variables were inserted simultaneously into the regression models for the relative risk (RR) of each variable was controlled for all other covariates. The model was initially adjusted by age and statistical significance was set at *p* < 0.05 and 95% confidence intervals (CI).

The observations without values were automatically removed by the statistical analysis software. All statistical analyses were conducted with Stata for Mac v.12 and statistical significance was established at *p* < 0.05 for all statistical tests.

### 2.4. Ethical Aspects

This study was carried out with anonymized public data; thus, the subjects could not be identified. According to Spanish legislation, an Ethics Committee is not required. Throughout this research project, we considered the ethical principles and recommendations of the American Psychological Association (2020), keeping the anonymity of the participants, the confidentiality of the data and a good research practice [35].

## 3. Results

### 3.1. Description of the Sample

The sample of 11,444 minors presented similar characteristics in terms of age, height, weight, and BMI (Table 1), except for the minors analyzed in 2001 and 2006 and those analyzed in 2001 and 2017, who showed significant differences in height and weight. In the specific analysis between boys and girls of different surveys, only the group of boys analyzed in 2001 and 2017 differed significantly in weight (*t*-test: *p* = 0.03).

Moreover, comparing separately the boys and girls of each edition of the survey, 2006 and 2017 were the only two years in which significant differences were detected between sexes in both weight and BMI. 

### 3.2. Prevalence of the Habits of PA and TV Viewing

The prevalence of children who practiced PA daily has decreased throughout the 20 years of the study period (Figure 1). The separate analysis of the three age groups showed a significant decrease of daily PA practice in all of them: between 6 and 10 percentage points, depending on the age group, with G2 and G3 presenting the sharpest and slightest decrease, respectively. Figure 1 shows that, in the three age groups, the comparison between the samples of consecutive surveys was significant in all cases.

When comparing the daily PA practice of each survey separately, in the first four surveys, the three age groups obtained significantly different results with respect to each other (*p* < 0.05). However, in the 2017 survey, the three subgroups showed similar behaviors in this variable (daily PA practice between 85% and 89%).

In contrast, the daily habit of watching TV has increased throughout the last two decades. Particularly, G1 and G2 showed an increase during the first analyzed decade, although it then decreased to one percent point below the level obtained in 1997 and to the same level of 1997, respectively. On the other hand, G3 presented significant increases in the percentage of children who watched TV daily both between 1997 and 2001, and between 2011 and 2017, with a final prevalence of three percentage points more compared to twenty years before. 

When comparing the habit of watching TV daily in each survey separately, statistically different behaviors were only detected between the three age groups in 2001 and 2017, in which G3 showed a habit of TV viewing of up to seven percentage points more than the younger groups (*p* < 0.01). On the other hand, the surveys of 1997, 2006, and 2011 showed a similar percentage of children who watched TV daily in the three age groups.

### 3.3. Frequency of PA Practice

Table 2 shows the frequency of PA practice over time according to gender. The proportion of children who “never” practiced PA almost doubled. However, this increase of sedentary lifestyle did not represent a decrease of children who practiced PA several times per week (i.e., “frequently”); in fact, there was an increase in the proportion of children who practiced PA “frequently”. On the contrary, the increase of sedentary lifestyle resulted from the decrease in the proportion of children who practiced PA “occasionally” and “regularly”.

In 2017, there were significant increases in PA practice in boys with respect to 2001 and 2011 (*p* < 0.001), as well as in girls in 2017 with respect to 2001, 2006, and 2011 (*p* < 0.001).

Likewise, after comparing the boys and girls of the same years, the two sexes showed significantly different behaviors with respect to each other in terms of PA practice in the five surveys (*p* < 0.001), with boys presenting a greater tendency to practice PA compared to girls. 

### 3.4. Hours of TV Viewing

Table 3 shows the number of hours of TV viewing over time and according to gender. As can be observed, the proportion of children who watched TV daily for more hours increased significantly (*p* < 0.05). Such increase was constant throughout the analyzed surveys, and the analysis by sex showed that the proportion of boys who watched TV for more than three hours per day increased three percentage points between 1997 and 2017, and that of girls increased fourfold (3.6% in 1997 vs. 13.1% in 2017). Moreover, it was observed that the obtained results were polarized: the children who watched TV between one and three hours per day decreased, with an increase in the proportion of those who did not watch TV, those who watched TV for less than one hour per day and those who watched TV for more than three hours per day.

It is worth highlighting that, in the year 2006, although the question and the response options were exactly the same as in the rest of the editions, none of the respondents stated that the minor with whom they lived watched TV for more than two hours per day in average. 

Similarly, after comparing the boys and girls of the same years, the two sexes showed significantly different behaviors with respect to each other in terms of TV viewing (*p* < 0.05) in three surveys (1997, 2001, and 2017), with a greater initial tendency of boys to watch TV for more hours, which then reversed in the results of 2017, where the proportion of girls who watched TV for more than three hours per day was higher than that of boys.

### 3.5. Determinants of PA Practice

Table 4 describes the results of multinomial logistic regressions between the independent variables (gender, daily hours of TV viewing, and BMI) and the outcome variable (frequency of PA practice). The latter was significantly associated with gender, daily hours of TV viewing, and BMI. Girls were 0.81 times more likely to practice PA occasionally and 0.37 times more likely to practice it frequently. The children who watched TV for more than three hours per day were between 0.58 and 0.43 times more likely to practice PA. Finally, obese children were 0.14 times more likely to practice PA frequently. The three independent variables were significantly associated with PA practice (0.37 < RR > 0.98).

## 4. Discussion

The aim of this study was to analyze the evolution of the habits of PA practice and TV viewing in a cohort of twenty years of research (1997–2017) in Spanish minors, in general, and children, specifically, in order to determine whether there is a relationship between these two activities. After the analysis of the results, it can be confirmed that both habits changed throughout the last two decades and that they are correlated.

Regarding the habit of practicing PA daily, the results show that its prevalence has decreased in general. Previous studies indicate that sedentary behaviors are highly prevalent in children and adolescents, which would justify the increasing rates of overweight and obesity among minors of western and developed countries [36,37]. However, our findings are not in line with such assertions; although a higher proportion of sedentary minors was detected, none of the surveys in any of the subgroups by age or sex showed an increase in body weight or BMI among the respondents with respect to previous editions. These results are in agreement with those obtained by some recent studies that also identified decreases in overweight and obesity rates in Spanish minors [38,39]. Particularly, the oldest group of minors (12–14 years) showed the lowest rate of daily PA practice abandonment, although it also presented the lowest prevalence of such habit in the initial survey of 1997. This age group, which can be considered as “pre-adolescent”, has been previously identified as predisposed to a sedentary lifestyle and the abandonment of exercise [40,41]. Therefore, PA habits in children from nine years of age (G2) are “adolescenting” and increasingly imitating the behavior pattern of minors between 12 and 14 years of age. This would explain why the large differences detected between the age groups in the surveys of 1997 have decreased gradually down to a difference of four percent points in 2017. 

However, such increase of totally sedentary children did not result from a decrease of those who practiced PA the most (several times per week or “frequently”), which increased. This phenomenon toward lifestyle “polarization” in Spanish children has never been identified before in a reliable manner. A plausible explanation is that the efforts made in the last years by Spanish public institutions and health agents to promote healthy lifestyles (in terms of exercise and healthy diet) and prevent comorbidities associated with sedentary behaviors are finally showing positive results [42,43]. In this way, the information and sensitization campaigns could be persuading parents and legal guardians to promote the habitual practice of physical exercise, although there is still a large percentage of the population who have not included it in their daily habits. The latter aspect is of special interest in the specific analysis of girls, who showed significantly more sedentary patterns than boys throughout the entire study period of twenty years. This phenomenon is a constant finding in similar studies, where girls always show less active lifestyles than boys of the same age [39,44].

In contrast, the results show that the prevalence of the habit of watching TV daily has increased throughout the last two decades, especially in G3. The group of pre-adolescents was, also in this aspect, the one that presented the most sedentary behavior. Moreover, in the analysis based on sex, girls also showed the greatest increase in the number of hours of TV viewing, eventually surpassing the proportion of boys who watched TV daily for the largest number of hours. This phenomenon contradicts some previous studies that identified higher rates of exposure to screens in boys compared to girls [18,39,45]. Specifically, Keane et al. [45] identified that the proportion of boys who spend more than one hour watching television daily is 3.3% higher than that of girls and those who spend more than one hour playing on the computer daily is 16.1% higher than that of girls. In parallel, Boente-Antela et al. [39] indicated that boys spend, on average, 40 more minutes per week in front of a screen than girls. In this sense, our results could be influenced by the fact that we did not consider the exposure of children to other electronic devices associated with a screen (such as game consoles, tablets, computers, and mobile phones), whose use is very common, especially in the last age group analyzed. In any case, the data do reveal the higher prevalence of the habit of watching TV for more hours per day in girls, although boys may be exposed for even more hours to other types of screens).

The logistic regression analysis allowed identifying the magnitude of the influence of the different study variables on the frequency of PA practice. Consequently, the variable that explains the greatest proportion of the variance of such variable is BMI. This relationship between both variables is reasonable, since the children with greater BMI are expected to be the ones who practice PA the least [18,46]. Then, the next variable with greater association identified was the number of daily hours of TV viewing; therefore, this habit can be interpreted as a “competitor” for the spare time of the children, making the greater exposure to this type of screen become a barrier for children to practice more PA. However, this assertion must be specified taking into account that, if there are barriers around the child that hinder his/her participation in activities associated with physical exercise (such as a lack of parental support, absence of close models of healthy lifestyle, lack of a safe environment where PA can be practiced in his/her community, etc.) [47,48], the child will carry out activities related to sedentary behaviors in his/her spare time, such as, in this case, watching TV.

Lastly, the variable “sex” also showed to be a predictor of the frequency of PA practice, being significantly less probably for a girl to practice PA several times per week compared to a boy. Although it presented the lowest predictive power, this variable has been confirmed by previous studies, which report that the differences between sexes in game style and spare time investment are significant from pre-school. These differences are strongly associated with stereotyped parental and social determinants (family economic level, parents’ educational level, or socioeconomic status) [49,50].

This study has important limitations that must be pointed out. Firstly, the information about weight, height, and BMI was self-reported; thus, it was not based on measurements made by an expert. Secondly, the exclusively Spanish population limits the generalization of our results to other populations. Thirdly, the habit of PA practice was also self-reported and categorized by the adult respondent who lived with the minor, without an objective quantification using a pedometer or accelerometer to measure the quantity, frequency, and intensity of the PA practiced. Lastly, the number of hours of TV viewing per day was self-reported, and thus was not measured objectively; therefore, it may not be a reliable measure of the sedentary habits of children, especially in the last two surveys, where the time of exposure to other types of screens and multimedia devices was not considered. Lastly, social aspects such as family economic level, parents’ educational level, or socioeconomic status have not been taken into account.

Despite these limitations, this study also has important strengths. This is a large representative study of the Spanish child population and, therefore a real reflection of the current situation of the habits of PA practice of the new generations. At the same time, this is the first study to detect direct relationships between the habits of PA practice and TV viewing in a cohort of 20 years of research of the Spanish population, with variables and correlations that behave in a similar manner as those of other populations, although with particularities for the Spanish population in other cases.

## 5. Conclusions

Since 1997, the number of Spanish children who practice PA daily has decreased significantly. Moreover, the number of children who watch TV daily for more than three hours per day has increased considerably. Particularly, pre-adolescents between 12 and 14 years of age and girls from 6 years of age are more likely to adopt more sedentary lifestyles. Furthermore, it was detected that the changes in both habits are related and that the number of hours that children spend watching TV influences the amount of PA they practice.

Therefore, the public health policies and initiatives oriented toward the reduction of sedentary behaviors and the promotion of healthy lifestyles must consider the differences between sexes, which are present already from childhood, in order for such interventions to be effective in the population of pre-adolescents, in general, and girls, in particular.

## Figures and Tables

**Figure 1 ijerph-17-06836-f001:**
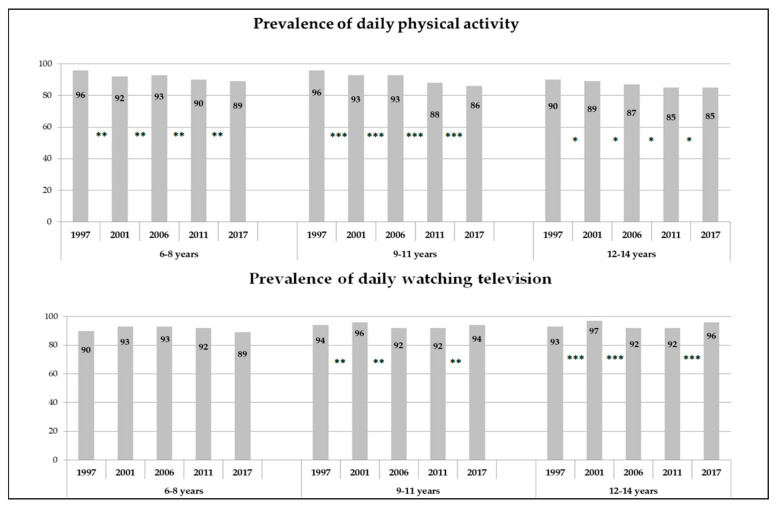
Prevalence evolutions of the habits of physical activity and television viewing (data provided: percentage) (Asterisks represent the ANOVA results between two consecutive surveys: * *p* < 0.05; ** *p* < 0.01; *** *p* < 0.001).

**Table 1 ijerph-17-06836-t001:** Descriptive statistics of the sample (data provided: mean ± standard deviation).

Survey		*N*	Age	Height	Weight	Body Mass Index
**1997**						
	Boys	423	10.5 ± 2.6	144.6 ± 18.5	40.9 ± 12.8	19.5 ± 8.5
	Girls	412	10.7 ± 2.5	146 ± 17.5	40.9 ± 11.7	19.2 ± 6.8
	All	835	10.6 ± 2.5	145.3 ± 18	40.9 ± 12.3	19.4 ± 7.7
**2001**						
	Boys	1131	10.3 ± 2.5	144.6 ± 17.3 ^a^	41 ± 13.2	19.2 ± 3.6
	Girls	1084	10.3 ± 2.5	143.6 ± 17.4	39.4 ± 12 ^b^	19.2 ± 10.7
	All	2215	10.3 ± 2.5	144.1 ± 17.3 ^a,b^	40.2 ± 12.6 ^a,b^	19.2 ± 7.9
**2006**						
	Boys	1969	10.5 ± 2.6	146.7 ± 18.2 a	42.4 ± 14.7 *	19.2 ± 4 *
	Girls	1920	10.4 ± 2.6	144.5 ± 16.7	40.3 ± 12.8 *	18.9 ± 3.8 *
	All	3889	10.4 ± 2.6	145.6 ± 17.5 ^a^	41.4 ± 13.8 ^a^	19.1 ± 3.9
**2011**						
	Boys	956	10.3 ± 2.6	145.1 ± 17.6	41.2 ± 13.8 *	19.2 ± 3.9 *
	Girls	780	10.3 ± 2.6	143.4 ± 16.7	39.7 ± 12.4 *	19 ± 3.7 *
	All	1736	10.3 ± 2.6	144.3 ± 17.3	40.5 ± 13.2	19.1 ± 3.8
**2017**						
	Boys	1405	10.4 ± 2.5	145.7 ± 18.1	41.5 ± 14.5	19.1 ± 4.2
	Girls	1364	10.6 ± 2.6	145.4 ± 16.9	41.2 ± 13.2 ^b^	19 ± 3.7
	All	2769	10.5 ± 2.6	145.6 ± 17.5 ^b^	41.3 ± 13.9 ^b^	19.1 ± 4

ANOVA results 2001 vs. 2006: ^a^
*p* < 0.05; ANOVA results 2001 vs. 2017: ^b^
*p* < 0.05; *t*-test between gender: * *p* < 0.05.

**Table 2 ijerph-17-06836-t002:** Frequency of physical activity practice [data provided: *n* (%)].

Survey	Never	Occasionally	Regularly	Frequently	All
**1997**					
Boys	19 (4.5%)	100 (23.6%)	143 (33.8%)	161 (38.1%)	423 (100%) ^aa, ccc,^ ***
Girls	37 (9%)	147 (35.7%)	137 (33.3%)	91 (22.1%)	412 (100%) ^ccc,^ ***
All	56 (6.7%)	247 (29.6)	280 (33.5%)	252 (30.2)	835(100%) ^aaa, b, ccc^
**2001**					
Boys	71 (6.3%)	358 (31.7%)	381 (33.7%)	321 (28.4%)	1131 (100%) ^aa, d, eee,^ ***
Girls	132 (12.2%)	439 (40.5%)	303 (28%)	210 (19.4%)	1084 (100%) ^eee,^ ***
All	203 (9.2%)	797 (36%)	684 (30.9)	531 (24)	2215 (100%) ^aaa, eee^
**2006**					
Boys	145 (7.4%)	503 (25.6%)	616 (31.3%)	705 (35.8%)	1969 (100%) ^d, ff,^ ***
Girls	222 (11.6%)	778 (40.5%)	526 (27.4%)	394 (20.5%)	1920 (100%) ^ff, ggg,^ ***
All	367 (9.4%)	1281 (32.9%)	1142 (29.4%)	1099 (28.3%)	3889 (100%) ^b, fff, ggg^
**2011**					
Boys	85 (8.9%)	257 (26.9%)	374 (39.1%)	240 (25.1%)	956 (100%) ^ccc, ff, hhh,^ ***
Girls	134 (17.2%)	298 (38.2%)	231 (29.6%)	117 (15%)	780 (100%) ^ccc, ff, hhh,^ ***
All	219 (12.6%)	555 (32%)	605 (34.9%)	357 (20.6%)	1736 (100%) ^ccc, fff, hhh^
**2017**					
Boys	141 (10%)	257 (18.3%)	451 (32.1%)	556 (39.6%)	1405 (100%) ^eee, hhh,^ ***
Girls	239 (17.5%)	321 (23.5%)	387 (28.4%)	417 (30.6%)	1364 (100%) ^eee, ggg, hhh,^ ***
All	380 (13.7%)	578 (20.9%)	838 (30.3%)	973 (35.1%)	2769 (100%) ^eee, ggg, hhh^

Chi^2^ results 1997 vs. 2001: ^aa^
*p* < 0.01; ^aaa^
*p* < 0.001; Chi^2^ results 1997 vs. 2006: ^b^
*p* < 0.05; Chi^2^ results 1997 vs. 2011: ^ccc^
*p* < 0.001; Chi^2^ results 2001 vs. 2006: ^d^
*p* < 0.05; Chi^2^ results 2001 vs. 2017: ^eee^
*p* < 0.001; Chi^2^ results 2006 vs. 2011: ^ff^
*p* < 0.01; ^fff^
*p* < 0.001; Chi^2^ results 2006 vs. 2017: ^ggg^
*p* < 0.001; Chi^2^ results 2011 vs. 2017: ^hhh^
*p* < 0.001; *t*-test between gender: *** *p* < 0.001.

**Table 3 ijerph-17-06836-t003:** Hours of watching television daily (*n* (%)).

Survey	Never	Less than One Hour	Between One and Two Hours	Between Two and Three Hours	More than Three Hours	All
**1997**						
Boys	19 (4.5%)	82 (19.4%)	195 (46.1%)	91 (21.5%)	36 (8.5%)	423 (100%) ^bbb, ccc, dd,^ ***
Girls	40 (9.7%)	96 (23.3%)	181 (43.9%)	80 (19.4%)	15 (3.6%)	412 (100%) ^aa, bbb, cc,^ ***
All	59 (7.1%)	178 (21.3%)	376 (45%)	171 (20.5%)	51 (6.1%)	835 (100%) ^aa, bbb, ccc, d^
**2001**						
Boys	41 (3.6%)	185 (16.4%)	552 (48.8%)	254 (22.5%)	99 (8.8%)	1131 (100%) ^eee, fff, ggg,^ ***
Girls	59 (5.4%)	212 (19.6%)	519 (47.9%)	223 (20.6%)	71 (6.6%)	1084 (100%) ^aa, fff,^ ***
All	100 (4.5%)	397 (17.9%)	1071 (48.4%)	477 (21.5%)	170 (7.7%)	2215 (100%) ^aa, eee, fff, ggg^
**2006**						
Boys	154 (7.8%)	494 (25.1%)	1321 (67.1%)	0 (0%)	0 (0%)	1969 (100%) ^bbb, eee, hhh^
Girls	153 (8%)	432 (22.5%)	1335 (69.5%)	0 (0%)	0 (0%)	1920 (100%) ^bbb, hhh^
All	307 (7.9%)	926 (23.8%)	2656 (68.3%)	0 (0%)	0 (0%)	3889 (100%) ^bbb, eee, hhh^
**2011**						
Boys	79 (8.3%)	412 (43.1%)	301 (31.5%)	78 (8.2%)	86 (9%)	956 (100%) ^ccc, fff, iii^
Girls	61 (7.8%)	349 (44.7%)	246 (31.5%)	65 (8.3%)	59 (7.6)	780 (100%) ^cc, fff, iii^
All	140 (8.1%)	761 (43.8%)	547 (31.5%)	143 (8.2%)	145 (8.4%)	1736 (100%) ^ccc, fff, iii^
**2017**						
Boys	78 (5.6%)	486 (34.6%)	498 (35.4%)	179 (12.7%)	164 (11.7%)	1405 (100%) ^dd, ggg, hhh, iii,^ *
Girls	101 (7.4%)	538 (39.4%)	412 (30.2%)	135 (9.9%)	178 (13.1%)	1364 (100%) ^ggg, hhh, iii,^ *
All	179 (6.5%)	1024 (37%)	910 (32.9%)	314 (11.3%)	342 (12.4)	2769 (100%) ^d, ggg, hhh, iii^

Chi^2^ results 1997 vs. 2001: ^aa^
*p* < 0.01; Chi^2^ results 1997 vs. 2006: ^bbb^
*p* < 0.001; Chi^2^ results 1997 vs. 2011: ^cc^
*p* < 0.01; ^ccc^
*p* < 0.001; Chi^2^ results 1997 vs. 2017: ^d^
*p* < 0.05; ^dd^
*p* < 0.01; Chi^2^ results 2001 vs. 2006: ^eee^
*p* < 0.001; Chi^2^ results 2001 vs. 2011: ^fff^
*p* < 0.001; Chi^2^ results 2001 vs. 2017: ^ggg^
*p* < 0.001; Chi^2^ results 2006 vs. 2017: ^hhh^
*p* < 0.001; Chi^2^ results 2011 vs. 2017: ^iii^
*p* < 0.001; *t*-test between gender: * *p* < 0.05; *** *p* < 0.001.

**Table 4 ijerph-17-06836-t004:** Multinomial Logistic Regression of physical activity practice in relation to gender, hour of watching television, and Body Mass Index, adjusted by age.

Variable	Occasionally		Regularly		Frequently	
	RR	95% CI	RR	95% CI	RR	95% CI
**Gender**						
Boy	1		1		1	
Girl	0.81 **	(0.71–0.93)	0.48 ***	(0.42–0.55)	0.37 ***	(0.32–0.42)
**Hours of watching television daily**						
Never	1		1		1	
Less than one hour	1.19	(0.89–1.58)	1.27	(0.96–1.68)	1.24	(0.94–1.64)
Between 1–2 h	1.30	(1.00–1.71)	1.04	(0.78–1.35)	0.96	(0.73–1.25)
Between 2–3 h	1.10	(0.79–1.52)	0.93	(0.68–1.29)	0.79	(0.57–1.09)
More than three hours	0.53 ***	(0.38–0.75)	0.58 **	(0.38–0.75)	0.43 ***	(031–0.60)
**Body Mass Index**						
Underweight	1		1		1	
Normal	0.95 *	(0.82–1.10)	0.86 *	(0.74–0.99)	0.79 **	(0.68–0.91)
Overweight	0.50 ***	(0.38–0.65)	0.45 ***	(0.34–0.58)	0.35 ***	(0.26–0.46)
Obese	0.52 **	(0.33–0.79)	0.29 ***	(0.18–0.47)	0.14 ***	(0.08–0.26)
**Gender**	0.81 **	(0.71–0.93)	0.48 ***	(0.42–0.55)	0.37 ***	(0.32–0.42)
**Hours of watching television daily**	0.87 ***	(0.81–0.94)	0.83 ***	(0.77–0.89)	0.77 ***	(0.72–0.83)
**Body Mass Index**	0.98 *	(0.97–0.99)	0.97 **	(0.96–0.99)	0.94 ***	(0.93–0.96)

The base outcome is “Does not practice physical activity”, * *p* < 0.05; ** *p* < 0.01; *** *p* < 0.001.

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
