# Peer review of "Evolution of the Habits of Physical Activity and Television Viewing in Spanish Children and Pre-Adolescents between 1997 and 2017"

_ijerph, 2020, doi:10.3390/ijerph17186836_

Round 1

Reviewer 1 Report

1.       The reliability and validity of the anthropometric measurements stated by parents in line 101 will add value to the manuscript.

2.       The content, face and construct validity of the questionnaire used in each survey will interest some readers line 88-99.

3.       In line 108-109- I suggest that the author  briefly provide the rationale for dividing the specific age group of children they have mentioned

4.       I suggest the author indicate the post hoc test they used for significant difference  in line 110 -111.

5.       I suggest the ethical clearance number be mentioned in line 124

6.       It is not clear how the authors accounted for cohort and period of measurements effects in the analysis.

7.       The authors in table 4 divided the hours of watching TV into four/five categories. The rational for this categories will interest some readers.

8.       One wonders if socioeconomic status of the population was considered.

Author Response

Dear Editor and Reviewer of International Journal of Environmental Research and Public Health:

Thank you very much for your suggestions and contributions to improve the quality of the manuscript. Following your indications, we respond, point by point, to the reviewers' comments.

In the text, all the modified or added sentences have been written in red to facilitate the correction by the reviewers.

  1. The reliability and validity of the anthropometric measurements stated by parents in line 101 will add value to the manuscript.

The authors have added this aspect and a bibliographic reference that corroborates it.

  1. The content, face and construct validity of the questionnaire used in each survey will interest some readers line 88-99.

The construct validity of the questionnaire used in each survey is information that the National Institute of Statistics of Spain does not provide. Instead, we have added the legal requirements assumed by the participants to participate and the reference to the official website of the Institute where you can find all the methodological information of the National Health Surveys.

  1. In line 108-109- I suggest that the author  briefly provide the rationale for dividing the specific age group of children they have mentioned.

The explanation to this methodological aspect has been added.

  1. I suggest the author indicate the post hoc test they used for significant difference in line 110 -111.

The Statistical Analysis subsection has been expanded by adding the required information.

  1. I suggest the ethical clearance number be mentioned in line 124.

According to Spanish and European legislation, this research does not require the approval of any Ethics Committee. This study was carried out with anonymized public data; thus the subjects cannot be identified.

  1. It is not clear how the authors accounted for cohort and period of measurements effects in the analysis.

The sample cohort was previously justified in the Material and Methods section, on the other hand, the effect of the measurement period has been established through all the statistical tests performed.

  1. The authors in Table 4 divided the hours of watching TV into four/five categories. The rational for this categories will interest some readers.

The division into five categories was established corresponding to the five existing answer options for that question in the surveys. This aspect is described in the Study variables subsection (the authors have marked it in red so that you can easily locate it, although it is not new or corrected content).

  1. One wonders if socioeconomic status of the population was considered.

No, the socioeconomic status of the population was not taken into account. The authors have added that aspect as a limitation of this research.

Once again, thank you very much for the time spent and the interest shown in this work; as well as in the positive evaluations you have given of it.

Receive a warm greeting,

The authors.

Reviewer 2 Report

Dear authors,

Please find enclosed the comments.

Kind regards

Author Response

Dear Editor and Reviewer of International Journal of Environmental Research and Public Health:

Thank you very much for your suggestions and contributions to improve the quality of the manuscript. Following your indications, we respond, point by point, to the reviewers' comments.

In the text, all the modified or added sentences have been written in red to facilitate the correction by the reviewers.

  1. Abstract: What is the conclusion or contribution of the work? The abstract ends with the results.

The authors have added the Conclusions subsection to the Abstract.

  1. Introduction: Please, include the “displacement hypothesis” (Mutz, Roberts & van Vuuren, 1993), because these two behaviors are related. However, there are current controversy because an adolescent could be active (reach 60 minutes of daily MVPA) and sedentary (not reach 120 minutes of daily screen time) at the same time.

The introduction has been expanded with the content you have provided us.

  1. Sex variable is included for the first time in the hypothesis. Please, refer in the introduction to the sex variable and in the aim.

     The possibility that the behavior in relation to physical exercise and television viewing is different in boys and girls has been introduced before the hypothesis and has been added in the aim.

  1. Participants: there are a = highlighted in yellow.

The = highlighted has been removed.

  1. Measures and instruments: Questionnaires for both variables are not accurate and not validated in youth population. There are many others that are more extensive and accurate. On the other hand, the current TV time (for instance, for the 2017 sample) includes more devices such as tablet or mobile and that is not measured. Questionnaires are a limitation to this study.

     The authors acknowledge the shortcomings of the study. In order to use the 20-year study cohort and the large sample, the authors were obliged to use the questionnaires that they had applied at the National Institute of Statistics of Spain.

     The authors have described in detail the methodological process of the National Health Surveys of Spain and we have added all the necessary bibliographic references.

     In addition, in recognition of what you indicate, the authors have expanded the limitations section so that these methodological aspects are adequately reflected.

  1. Results: in several tables appears height, weight and BMI. These variables are interesting, however, there are not included in the introduction and the aim.

The authors have expanded the Introduction and, now, these variables are presented in the second paragraph of this section.

Once again, thank you very much for the time spent and the interest shown in this work; as well as in the positive evaluations you have given of it.

Receive a warm greeting,

The authors.

Reviewer 3 Report

Reviewer #2:

  1. Thank you for the opportunity to review this manuscript. Overall, this is an interesting study analysing the evolution of the habits of physical activity practice and television watching in a cohort of 20 years of research in Spanish children. However, this article would benefit from using the Strengthening the Reporting of Observational Studies in Epidemiology (STROBE). I have some suggestions and comments below for the authors to consider:

General suggestions:

  1. Check the use of correct verbal time along the text.
  2. Please correct the page numbering of the manuscript.
  3. The quality of the article would benefit from having an academic English teacher to review the manuscript.

 I provide major comments below for the authors' consideration:

Abstract:

  1. Page 1, lines 21-22 ‘Methods’: It would be interesting to include more information about the methods of this study, such as which obesity-related behaviours had been examined.
  2. Page 1, lines 23-27 ‘Results’: It would be useful to provide the results in numbers, such as the percentages.

Introduction:

  1. Page 1, lines 40-41: “In the specific study of the children population, the number of 5-year-old children with obesity are expected to be 41 million by the year 2030” – This sentence is unclear and requires a citation.
  2. Page 1, line 40-45: It would be useful to define "children” and “pre-adolescents”, emphasizing your target population.
  3. Page 2, lines 46-55: It would be useful to provide some epidemiological data about child physical activity in Spain.
  4. The hypothesis of the study were not identified. It would be interesting to include this. 

Methods:

  1. Page 3, line 78: How the personal interviews were conducted?
  2. Page 3, lines 89-94;97-99: Could the authors please provide some more information about the questionnaire used? It would be important to include the psychometric properties of the tool, such as the internal consistency of the scales.

Results:

  1. Figure 1: Please add the term ‘percentage’ in the left side of the figures.

 Discussion:

  1. In general, it would be interesting to discuss the findings of the other studies more specifically, such as “This phenomenon contradicts some previous studies that identified higher rates of exposure to screens in boys compared to girls [30,36, 37]” – Please provide details of these studies.
  2. Page 11, lines 321-322: “These differences are strongly associated with stereotyped parental and social determinants [42]” – Which social determinants are they?

Conclusions:
16. The conclusion is not clear. State the main results, the impact of the research, and then, the consideration for future studies. For example, “Since 1997, the number of children who practice PA daily has decreased significantly…” – Children from where? From your own study?

Author Response

Dear Editor and Reviewer of International Journal of Environmental Research and Public Health:

Thank you very much for your suggestions and contributions to improve the quality of the manuscript. Following your indications, we respond, point by point, to the reviewers' comments.

In the text, all the modified or added sentences have been written in red to facilitate the correction by the reviewers.

  1. General suggestions: check the use of correct verbal time along the text; please correct the page numbering of the manuscript; the quality of the article would benefit from having an academic English teacher to review the manuscript.

The numbering of the pages is automatic through the journal template. Since the time allowed by the journal to return the corrected manuscript is very limited, we have not been able to send the manuscript to an academic English teacher to review it. But the authors promise to send it to correction if it is accepted before it is published.

  1. Abstract:
  • Page 1, lines 21-22 ‘Methods’: It would be interesting to include more information about the methods of this study, such as which obesity-related behaviours had been examined.
  • Page 1, lines 23-27 ‘Results’: It would be useful to provide the results in numbers, such as the percentages.

     The abstract has been expanded with the dependent variables analyzed and the Results subsection has been modified by adding percentages.

  1. Introduction:
  • Page 1, lines 40-41: “In the specific study of the children population, the number of 5-year-old children with obesity are expected to be 41 million by the year 2030” – This sentence is unclear and requires a citation.
  • Page 1, line 40-45: It would be useful to define "children” and “pre-adolescents”, emphasizing your target population.
  • Page 2, lines 46-55: It would be useful to provide some epidemiological data about child physical activity in Spain.
  • The hypothesis of the study were not identified. It would be interesting to include this. 

     The authors have rewritten the phrase to which you refer and have added the bibliographic reference at the end of it.

     The differentiation between "children" and "pre-adolescents" has been added in Material and Methods in the Statistical analysis subsection to justify the division into age subgroups of the sample. We have added it in that section at the suggestion of Reviewer nº2.

     The authors have added the epidemiological data that it indicates to us and the research hypotheses.

  1. Methods:
  • Page 3, line 78: How the personal interviews were conducted?
  • Page 3, lines 89-94;97-99: Could the authors please provide some more information about the questionnaire used? It would be important to include the psychometric properties of the tool, such as the internal consistency of the scales.

That detail has been added: the interviews were conducted by telephone.

In the subsection Study variables, more information has been added about the methodology used in the National Health Surveys in Spain. However, you should bear in mind that it is not a validated questionnaire like those used in the scientific literature. It is a huge battery of questions related to very different aspects of lifestyle and health. In the Study variables subsection, the authors have reflected all the methodological aspects provided by the National Institute of Statistics of Spain (and we have referenced them).

  1. Results: Figure 1: Please add the term ‘percentage’ in the left side of the figures.

     The authors have added this detail at the end of the title (as it is a prevalence, it must always be considered as a percentage).

  1. Discussion:
  • In general, it would be interesting to discuss the findings of the other studies more specifically, such as “This phenomenon contradicts some previous studies that identified higher rates of exposure to screens in boys compared to girls [30,36, 37]” – Please provide details of these studies.
  • Page 11, lines 321-322: “These differences are strongly associated with stereotyped parental and social determinants [42]” – Which social determinants are they?

     The Discussion section has been expanded with the aspects that you advise us.

  1. Conclusions: The conclusion is not clear. State the main results, the impact of the research, and then, the consideration for future studies. For example, “Since 1997, the number of children who practice PA daily has decreased significantly…” – Children from where? From your own study?

     The conclusions have been rewritten.

Once again, thank you very much for the time spent and the interest shown in this work; as well as in the positive evaluations you have given of it.

Receive a warm greeting,

The authors.